# Infantile Brain Tumors: A Review of Literature and Future Perspectives

**DOI:** 10.3390/diagnostics11040670

**Published:** 2021-04-08

**Authors:** Valeria Simone, Daniela Rizzo, Alessandro Cocciolo, Anna Maria Caroleo, Andrea Carai, Angela Mastronuzzi, Assunta Tornesello

**Affiliations:** 1Pediatric Oncology Unit, Ospedale Vito Fazzi, Piazza Filippo Muratore, 1, 73100 Lecce, Italy; danielarizzo980@gmail.com (D.R.); acocciolo@live.it (A.C.); 2Department of Onco-Hematology and Cell and Gene Therapy, Bambino Gesù Children’s Hospital (IRCCS), Piazza Sant’Onofrio 4, 00146 Rome, Italy; annamaria.caroleo@opbg.net (A.M.C.); angela.mastronuzzi@opbg.net (A.M.); 3Neurosurgery Unit, Department of Neurological and Psychiatric Sciences, Bambino Gesù Children’s Hospital (IRCCS), Piazza Sant’Onofrio 4, 00146 Rome, Italy; andrea.carai@opbg.net

**Keywords:** brain tumors, tumors in infants, congenital cancer, neuro-oncology, neonatal cancer

## Abstract

Brain tumors in infants including those diagnosed in fetal age, newborns and under a year old represent less than 10% of pediatric nervous system tumors and present differently when compared with older children in terms of clinical traits, location and histology. The most frequent clinical finding is a macrocephaly but non-specific symptoms can also be associated. The prognosis is usually poor and depends on several factors. Surgery continues to be the main option in terms of therapeutic strategies whereas the role of chemotherapy is not yet well defined and radiotherapy is exceptionally undertaken. In view of this situation, a molecular characterization could assist in providing therapeutic options for these tumors. This review highlights the recent advances in the diagnosis and treatment of brain tumors in infants with a particular focus on the molecular landscape and future clinical applications.

## 1. Introduction

Brain tumors arising in children under a year old are a rare subgroup of central nervous system (CNS) neoplasms that can pose a difficult challenge to the neuro-oncologist. Their diagnosis is often delayed because symptoms, due to the mobility of the skull bones, can occur late. Furthermore, they demonstrate a biologically more aggressive behavior when compared with the same histological entities in older children [1]. Their treatment is also complicated by the fact that there are few therapeutic weapons available and all come with potential risks: surgery has related operative-anesthesiological risks; chemotherapy and radiotherapy can have long-term complications [2]. Although their molecular characterization is still limited, this could represent an important element in therapeutic perspectives. With this in mind, infantile brain tumors should be considered as a distinct entity in the broader landscape of pediatric brain tumors with their own diagnostic and therapeutic algorithms following what has already been done in the classification of other malignancies such as acute lymphoblastic leukemia [3]. The aim of this mini-review is to summarize the current knowledge of infantile brain tumors and take an in-depth look at recent advances in their diagnosis and treatment.

## 2. Materials and Methods

The authors examined the literature available on infant brain tumors. Research studies were selected based on research topics. The search terms used were congenital brain tumors, neonatal brain tumors, infantile brain tumors and fetal brain tumors. The search was conducted on the PubMed database from 1980 to 2020. These research studies were classified according to their relevance. The information found in the selected studies was carefully evaluated with particular attention given to epidemiology, histological and biological characteristics, symptoms, diagnosis and treatment. They are described and discussed in the following sections. A few of the main studies in the literature regarding infantile brain tumors are summarized in Table 1.

## 3. Definition and Epidemiology

Malignant brain tumors are the most common solid malignancies in childhood, accounting for approximately 16–23% of all tumors [14]. They are the second most common pediatric malignancy after leukemias. CNS tumors in infants are a rare entity and are defined as brain tumors occurring in children less than one year of age and thus include tumors diagnosed in fetal age, neonatal age and under the age of one year. Infantile brain tumors constitute approximately 10% of all pediatric CNS tumors and about half occur in the first six months of life [15,16]. Congenital brain tumors still lack a clear definition in terms of being a subgroup and have been divided by a few authors into “definitely”, “probably” and “possibly” congenital depending on whether they occur at birth, within the first week of life or within the first six months of life [17]. This definition as ‘congenital’, although not yet universally accepted, seems reasonable and clinically relevant, particularly as the diagnosis in the fetal age is increasing due to better prenatal imaging techniques [18]. In addition, it cannot be ruled out that there are a number of unaccounted for fetal age brain tumors due to miscarriage or abortion [5].

## 4. Clinical Findings

The most common symptoms of a brain tumor in the pediatric population are headaches and early morning vomiting. In infant patients, we must rely on more subtle and less specific signs and symptoms. The fontanelles allow for the stretching and deformation of the head as the brain expands faster than the surrounding bone. This flexibility often leads to a delay in the onset of symptoms as the infant skull is capable of accommodating the increase in intracranial pressure and when symptoms do arise they are often non-specific. These patients may simply appear sleepy and irritable. Additionally, specific symptoms may be present depending on the tumor site [19,20].

Prenatal signs: in the antenatal period, polyhydramnios, secondary to depressed swallowing from a hypothalamic dysfunction, is the most common feature and is present in approximately one third of patients. It may be the first clinical indication noted during the obstetric examination [21]. Another common feature is macrocephaly that can either be a consequence of the intracranial expansion of the tumor or of hydrocephalus. Hydrocephalus is generally caused by the obstruction of the ventricular system but it can also arise from increased cerebrospinal fluid production from a choroid plexus tumor. These anomalies can be detected by a prenatal ultrasound and they are most commonly encountered in the third trimester [8]. In these cases, fetal magnetic resonance imaging (MRI) can help to confirm these findings but can rarely help differentiate between individual tumor types. Some of these tumors such as intracranial teratomas, glioblastomas and primary neuroectodermal tumors (PNETs) can grow significantly during gestation and cause stillbirth [22]. Brain tumors diagnosed during gestation often require a cesarean section because of the associated complications mentioned. Neonates born by vaginal delivery often develop dystocia because of large fetal skulls. Wakai et al. observed in 115 cases of congenital cancer a 32% dystocia at delivery; 35 babies (30.4%) were stillborn, while 28 (24.3%) were premature [5].

Postnatal signs: macrocephaly could also be the first sign of a brain tumor in the postnatal period. This may be associated with a delayed fusion of the anterior fontanelle (normally fused by approximately 12 months) or bulging fontanelles [23]. Later in infancy, other symptoms may arise such as a failure to thrive, apneic episodes, irritability, a delay in developmental milestones, drowsiness, irritability, seizures, somnolence, vomiting and abnormal eye movements. These symptoms are non-specific and often difficult to detect upon clinical examination of the newborn. As emerged from the retrospective cohort of Toescu et al., the most common presenting symptom in the first year of life is vomiting and the most common clinical sign is macrocephaly followed by a bulging fontanelle [13].

Congenital anomalies such as cleft lip, cleft palate, heart and urinary tract malformations sometimes may suggest the presence of associated congenital tumors. Cleft palate/lip is associated with teratomas and low-set ears are associated with craniopharyngiomas. The identification of a brain tumor and an intracardiac mass such as a rhabdomyoma at times could suggest a genetic syndrome such as the tuberous sclerosis complex (TSC), implying that a diagnostic approach including genetic counseling is needed [24].

The primary differential diagnosis for an intracranial tumor in infants is hemorrhage, which may also manifest as a disorganized intracranial mass and/or hydrocephalus. Conversely, congenital CNS tumors have a propensity to bleed intratumorally. Therefore, when there is evidence of a spontaneous intracranial hemorrhage, an underlying neoplasm should always be excluded [17].

## 5. Diagnosis

A histopathological examination is essential for a definitive diagnosis of a brain tumor but they can be exceedingly difficult to obtain in infants and it is virtually impossible to safely obtain a fetal biopsy specimen [8]. Therefore, imaging has a fundamental role in a diagnosis.

Cranial ultrasonography (US) and MRI are the mainstays of diagnostic evaluation. During ultrasonography, most intracranial tumors have a heterogeneous pattern with the subversion of normal structures. In particular, teratomas are usually associated with calcifications, which may be important for the diagnosis. When the suspicion of a tumor arises, an MRI should be the next step. This technique allows for a detailed assessment of the tumor morphology and its spatial relationships with the surrounding structures; essential information for an eventual surgical approach. A disadvantage of MRI is its inadequacy in identifying the calcifications that are characteristic of a few histological subtypes such as oligodendrogliomas and gangliogliomas. Computed tomography (CT) is more adequate in achieving this aim but exposes the patient to a large dose of ionizing radiation [25]. Another disadvantage of MRI is that it requires time for acquiring images and the patient must remain perfectly still throughout the scan. This difficulty can be overcome with sedation but this obviously exposes the baby to an anesthetic risk. New strategies to immobilize the child without anesthesia are being explored such as performing the examination immediately after a meal, at the peak of drowsiness or using infant incubators/immobilizers or sucrose solutions [26].

## 6. Tumor Subtypes

As already discussed, obtaining a biopsy specimen is extremely difficult in infants therefore the histopathologic diagnosis is conducted, in most cases, after the tumor is surgically resected for therapeutic reasons. The gestational age may be helpful in finding a specific diagnosis of congenital CNS tumors. Teratomas and hamartomas generally develop before 22 weeks, germ cell tumors between 22 and 32 weeks and astrocytomas and glioblastomas after 32 weeks [5]. The distribution of histological subtypes in children < one year old is different from that of older children. Teratoma is the most frequent congenital tumor (approximately one third to one half of all cases), followed by gliomas and choroid plexus papillomas (18–47% and 5–20% of all perinatal brain tumors). Embryonal tumors including medulloblastomas and atypical teratoid/rhabdoid tumors (ATRTs) are less frequent. Ependymoma is another subtype diagnosed in infants with a higher incidence under four years. Pinealoblastomas and craniopharyngiomas are other tumors encountered more rarely in this age group [27,28]. The information on the molecular characterization of childhood brain tumors is still extremely limited (Figure 1). This constitutes a major problem also on the therapeutic level, lacking elements that can guide the definition of an algorithm for the evaluation and management of these types of tumors. Epigenetics also represent a central focus in pediatric brain tumor pathogenesis. Several studies have revealed that a specific miRNA signature or DNA methylation profile could help distinguish a tumor subgroup with a consequent peculiar therapeutic approach in pediatric brain tumors [29,30]. A specific miRNA expression has been observed in an atypical teratoid/rhabdoid tumor (ATRT), an ependymoma, a glioblastoma, a medulloblastoma and a pilocytic astrocytoma [31]. However, several aspects still need to be clarified such as the lack of correlation between miRNA expression levels in sera and in tumor tissues [32]. The existence of specific epigenetic features in infants is still unknown but it could represent an important field of study for its diagnostic and therapeutic implications.

### 6.1. Perinatal Teratoma

A perinatal teratoma generally involves the pineal or suprasellar region; however, cases of teratomas affecting cerebellar vermis, lateral ventricles or basal ganglia are known. Both mature and immature subtypes have been diagnosed in the perinatal setting. “Adult-type” tissue elements characterize the mature teratoma while immature elements such as primitive neuroectodermal tissues with multilayered rosettes would suggest a histological diagnosis of an immature teratoma. Although a teratoma may present as a massive tumor with a poor outcome, until now there have been no cases of a malignant transformation described in the perinatal setting [33].

### 6.2. Glial Tumors

Glial tumors are more often low-grade but high-grade gliomas, distinctively driven by gene fusions, may occur rarely. B-Raf proto-oncogene, serine/threonine kinase (BRAF) V600E mutation is a molecular finding observed in almost 20% of patients with pediatric low-grade gliomas (LGGs) including infants. BRAF V600E has been associated with a worse outcome compared with patients with the BRAF wild-type especially when there is Cyclin-Dependent Kinase Inhibitor 2A (CDKN2A) deletion [34]. Consequently, the use of BRAF inhibitors has already demonstrated a clinical benefit both for cytoreduction and the control of clinical symptoms [35].

Among low-grade gliomas, subependymal giant cell astrocytomas (SEGAs) are commonly diagnosed at or before birth and are strongly linked to tuberous sclerosis [36]. SEGAs are typically located near the foramen of Monro in the lateral wall of the lateral ventricles and have peculiar histological features. Large polygonal cells with abundant eosinophilic cytoplasms, sometimes with pleomorphic nuclei, multinucleated cells and spindle cells with a variable expression of glial fibrillary acidic protein (GFAP) associated with a smaller proportion of dysmorphic neuronal cells, characterize the histopathology of SEGAs. Surgical resection and targeted medical therapy with mTOR inhibitors such as an everolimus are the main therapeutic approaches for SEGAs. Surgical mortality and morbidity are estimated at 1–25% and 5–50%, respectively, but surgery is the first choice for high volume tumors and to control the hydrocephalus [37]. An everolimus has a response rate of about 60% and can spare the patient from surgery [38].

### 6.3. Choroid Plexus Tumors

Perinatal and infantile cases of choroid plexus tumors are preferentially located in the lateral ventricles. Typical and atypical (> 2 mitoses/10HPF) choroid plexus papillomas as well as rare malignant examples, i.e., a choroid plexus carcinoma, have been described [39]. In most cases, the diagnosis is supported by a morphology with delicate papillae covered by a monolayer of columnar cells and a characteristic immunophenotype (CK20+, CK7−, KIR7.1+). Despite their indolent course, survival from this type of tumor in infancy is very low also because of their marked bleeding propensity.

### 6.4. Medulloblastomas

Infantile medulloblastomas generally fall in the Sonic Hedgehog (SHH) -activated TP53-wild-type molecular subgroup and belong to the desmoplastic/nodular (DN) histological subgroup or to its closely related variant, i.e., medulloblastoma with extensive nodularity (MBEN) [40]. Morphologically, the DN group is characterized by pale, reticulin-free nodules composed of variously mature neurocytic cells that are embedded in a fibrillar matrix with a low proliferative activity and are surrounded by highly proliferative atypical cells embedded in a desmoplastic intercellular stroma. In the MBEN variant, the reticulin-free nodules are dominant and tend to coalesce together in an irregular way [41]. Considering the molecular subgroups of SHH, medulloblastomas diagnosed in infants fall in the SHH β and SHH γ, with a consequently different outcome. SHH β is characterized by a higher rate of metastatic dissemination that has been associated with a higher number of PTEN deletions and focal amplifications. The SHH γ group is characterized by a higher incidence of the MBEN variant that has an indolent behavior [42]. Thus, the identification of a SHH γ subtype stratifies the infant in a low-risk group that may not require a high dose chemotherapy treatment with consequent benefits in terms of long-term toxicity [43]. The activation of the SHH pathway in the MBEN variant has been associated with germline and/or somatic inactivating mutations of SUFU or PTCH1; the latter is associated with Gorlin Syndrome (GS) [44,45]. Compared with older children, the prognosis of an early childhood medulloblastoma is less satisfactory (five year OS < 70%) excluding SHH medulloblastomas [46]. This is related to the higher frequency of metastasis at the diagnosis and the limited use of radiotherapy to avoid long-term sequelae. Recently, several strategies based on surgery and chemotherapy in order to delay/avoid craniospinal radiotherapy have shown better survival rates and fewer long-term sequelae [47,48,49].

### 6.5. Ependymomas (EPN)

Ependymomas (EPN) in children usually occur in the posterior cranial fossa (PF) or at the supratentorial (ST) level [50]. An anaplastic (grade III) ependymoma is the histological type most frequently diagnosed in infants [51,52]. In the context of molecular groups identified by the study of DNA methylation patterns, the subgroups ST-EPN-RELA [53] and ST-EPN-YAP1 are the most frequent among the supratentorial ependymomas. They are characterized, respectively, by RELA and YAP 1 gene fusions. ST-EPN-RELA is associated with a poor outcome while ST-EPN-YAP1 is associated with a better outcome [50]. Another molecular subtype diagnosed in infants is PF-EPN-A, which is the most frequent of all ependymomas diagnosed in infancy [52]. The gain of chromosome 1q (1q+) in this tumor subgroup is related to an inferior outcome. In the same subgroup, the loss of p16 is also associated with a poor clinical outcome, however. The deregulation of the p16-CDK4/6-pRB-E2F pathway suggests a potential role for CDK4/CDK6 inhibitors [54]. Despite advances in understanding the biology of ependymomas, the first therapeutic approach remains a maximal safety surgery followed by chemotherapy [52]. Radiation therapy is indicated in patients > 12 months owing to the treatment related sequelae [51]. Infants with ependymomas face the worst prognosis in the disease setting [55].

### 6.6. Atypical Teratoid/Rhabdoid Tumor (ATRT)

ATRT is an aggressive embryonal tumor diagnosed mainly in children under three years old [56]. This tumor is triggered by the inactivation of SMARCB1 observed in the majority of cases or, rarely, SMARCA4. Three molecular subgroups have been identified [57]. ATRTs diagnosed under one year of age belong to the TYR molecular subgroup, characterized by an overexpression of the enzyme tyrosinase and by an infratentorial location in two thirds of cases [58]. Congenital cases have also been reported mainly in the context of rhabdoid tumor predisposition syndrome 1 or 2, associated with the germline mutation of SMARCB1 or SMARCA4, respectively [59]. The biallelic inactivation of SMARCB1 leads to genetic instability, the overexpression of cell cycle activators and tumorigenesis [60]. Morphologically ATRTs are heterogeneous consisting of large “plasmacytoid” cells and characteristic “rhabdoid” cells with eccentrically located nuclei and prominent nucleoli and a variable amount of eosinophilic cytoplasm sometimes containing globular inclusions [61]. The absence of the expression of SMARCB1/INI1 is a fundamental clue for the diagnosis. In cases that unexpectedly retain SMARCB1/INI1, it is important to investigate the SMARCA4/BRG1 expression because its inactivation predisposes a worse prognosis and the associated increased risk of a germline mutation [62]. A case of a recurrent rhabdoid tumor with a mutation in the cell cycle inhibitor CDKN1C in addition to the hallmark biallelic loss of SMARCB1 has recently been reported [63]. SMARCB1 regulates CDKN1C through LIN28B, a protein involved in early embryogenesis [64], which upregulates aurora kinase A (AURKA). Its activation contributes to genetic instability and favors tumor cell proliferation [65]. Based on recent reports of efficacy in children with recurrent rhabdoid tumors, AURKA inhibitors were considered as an adjuvant therapy for these patients [64]. Children diagnosed with ATRT have generally been seen to have a poor prognosis but recently the use of multimodal treatment strategies or high dose chemotherapy regimens have improved the overall survival [66,67].

### 6.7. Congenital Glioblastoma (cGBM)

A congenital glioblastoma (cGBM) is among the rarest types of congenital brain tumors. The prognosis of these tumors is extremely poor ranging from stillborn babies to children with a survival of less than two months if untreated. This unfavorable prognosis may in part be due to the tendency for bleeding and thus intracranial hemorrhage even though there are a few reports of patients with a good outcome after limited or no treatment [68,69]. These observations indicate that cGBMs may have a more unpredictable and perhaps a more favorable outcome than pediatric and adult cancers. This is likely attributable to the different molecular alterations that cGBMs have compared with tumors in older children and adults, as detailed by Ceglie et al. [3]. The evaluation of gene expression profiles led to the identification of 31 differentially expressed genes in congenital glioblastomas (cGBMs) compared with pediatric non-congenital glioblastomas (pGBMs) and primary adult GBMs (aGBMs) [70]. Mutations in TP53 are less common in children than in adults as well as PTEN deletions and EGFR mutations that are infrequent in children [71]. Similarly, no platelet-derived growth factor receptor alpha (PDGFRa) amplification was observed in cGBMs and PDGFRa gene expression levels were much lower than those seen in a group of patients with a pGBM [72]. Additionally, the upregulation of the MYCN proto-oncogene and the BHLH transcription factor (MYCN) did not characterize the gene expression data from a cGBM [70]. None of the patients with cGBMs had the recently described histone H3.3 mutations found in pGBMs, which are extremely rare in aGBMs [73]. Another molecular characteristic described in a pediatric high-grade glioma is represented by the identification of gene fusions involving the neurotrophin receptor tyrosine kinase genes 1–3 (NTRK1, NTRK2 and NTRK3), which encode for tropomyosin receptor kinase (TRK) A, B and C. This finding is observed in 40% of non-brainstem high-grade gliomas (HGGs) in infants [74,75]. These fusion genes lead to the transcription of chimeric TRK proteins that, in activating the kinase function, favor the tumorigenesis. Thus, new drugs with anti-TRK activity assume a potential role in the treatment of this tumor subtype [76]. These data support the idea that distinct molecular pathways are involved in the tumorigenesis of pediatric and adult GBMs.

## 7. Treatment and Prognosis

The principal treatment for infantile brain tumors is surgery, aiming for gross total resection. Surgical radicality is directly related to prognosis [77,78]. Moreover, surgery also offers histological specimens to allow for the most accurate diagnosis. Radiotherapy and chemotherapy are therapeutic strategies that can complement surgery in CNS tumors. Craniospinal irradiation was the main treatment for childhood CNS tumors but because of the severe sequelae (mental retardation, endocrine dysfunction, secondary neoplasms in the CNS) it is currently contraindicated under three years of age. In this age group, radiation therapy is also associated with a worse outcome as well as more severe side effects [25]. As radiation therapy is not an option in infants, the use of adjuvant chemotherapy and its effect on the outcome is being explored with patients with ATRTs, medulloblastomas, LGG and even infantile ependymoma [79].

The most common outcome of infantile brain tumors is death within five years of the diagnosis with overall survival rates around 30% [80] and the few survivors are likely to suffer long-term morbidity. Several studies have shown that virtually all children had neurological sequelae with residual neurological or developmental deficits at follow-up [2]. This is why a better molecular understanding of these neoplasms might lead to the identification of new therapeutic targets thus improving the current dismal prognosis of infantile brain tumors.

## 8. Conclusions

Infantile brain tumors are a rare entity in pediatric oncology and currently still lack a complete characterization. In this article, we have summarized the current knowledge of these tumors and provided an up-to-date review of the main characteristics paying particular attention to the molecular landscapes. It is known that these tumors behave more aggressively when compared with their pediatric and adult counterparts mainly because of their greater growth rate and the relatively small number of therapeutic options. In this population, surgery represents the main curative option; radiotherapy is out of the question because of the very young age of the patients and the role of chemotherapy for the same reason is still not well defined. In this context, it is extremely important to improve the molecular characterization in order to find new targets for the possible identification of new biology-driven therapeutic approaches. Our hope is that with a better understanding of the tumor biology, we will be able to improve the current poor prognosis of these children.

## Figures and Tables

**Figure 1 diagnostics-11-00670-f001:**
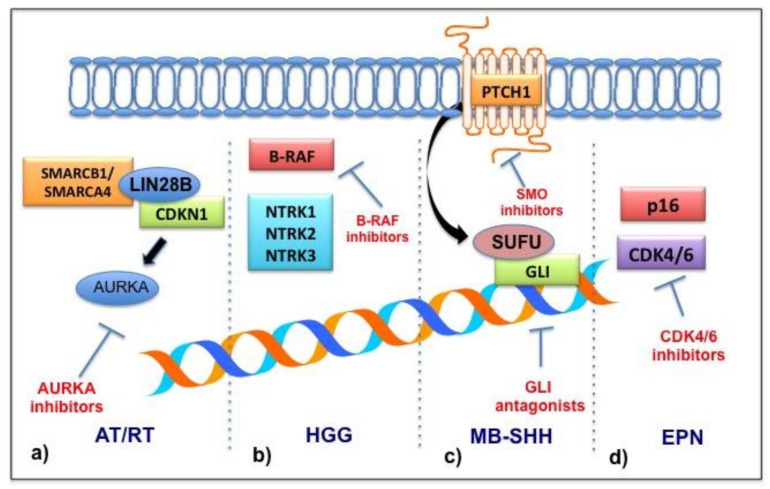
Molecular pathways and potential druggable targets in brain tumors of infants. (**a**) AT/RTs are characterized by the inactivation of SMARCB1 or SMARCA4. These molecules regulate cyclin dependent kinase inhibitor 1C (CDKN1C) through LIN28B that upregulates aurora kinase A (AURKA). (**b**) BRAF mutations have been detected in HGGs and LGG and BRAF inhibitors have proven effective. Gene fusions involving the neurotrophin receptor tyrosine kinase genes 1–3 (NTRK1, NTRK2 and NTRK3) have been identified in HGGs and new anti-TRK drugs are promising. (**c**) Inactivating mutations of PTCH1 or SUFU characterize the subgroup SHH of a medulloblastoma. As SUFU is a negative regulator of the glioma-associated oncogene (GLI) transcriptional factors that activate the Hedgehog (HH) pathway, GLI antagonists are new potential targeted drugs. (**d**) The loss of p16 in ependymoma leads to the activation of the p16-CDK4/6-pRB-E2F pathway, suggesting a potential role of CDK4/6 inhibitors. AT/TR: atypical teratoid rhabdoid tumor; HGG: high-grade glioma; MB-SHH: medulloblastoma, subgroup SHH; EPN: ependymoma.

**Table 1 diagnostics-11-00670-t001:** Retrospective studies or epidemiological surveys concerning intracranial tumors in infancy.

Reference	Number of Patients (or Biopsy Samples)	Age or Period of Life at Diagnosis	Most Represented Histotypes	Num. Ref.
Raimondi, 1983	39	0–1 year	Medulloblastoma Astrocytoma	[4]
Wakai, 1984	200	0–2 months	Teratoma Astrocytoma Medulloblastoma	[5]
Buetow, 1990	45	Diagnosis within 60 days after birth	Teratoma Astrocytoma PNET	[6]
Haddad, 1991	22	0–1 year	Astrocytoma PNET Choroid Plexus Tumor	[7]
Isaacs, 2002	250	Congenital (producing a sign or symptoms at or before birth)	Teratoma Astrocytoma Choroid Plexus Papilloma	[8]
Cassart, 2008	27	Fetal age (18–36 weeks)	Teratoma Glial Tumors Hamartoma	[9]
Qaddoumi, 2011	27	First 120 days of life	Glial Tumors AT/RT Ependymoma	[10]
Ghodsi, 2015	31	0–1 year	PNET Anaplastic Ependymoma Classic Ependymoma	[11]
Munjal, 2016	64	0–1 year	Low-Grade Glioma Germ Cell Tumor Choroid Plexus Papilloma	[12]
Toescu, 2018	98	0–1 year	Choroid Plexus Papilloma PNET AT/RT	[13]

^1^ AT/RT, atypical teratoma/rhabdoid tumors; PNET, primitive neuroectodermal tumors.

## Data Availability

Not applicable.

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
