# Peer review of "Infantile Brain Tumors: A Review of Literature and Future Perspectives"

_diagnostics, 2021, doi:10.3390/diagnostics11040670_

Round 1
Reviewer 1 Report
It should indicate the publication Infantile/Congenital High-Grade Gliomas: Molecular Features and Therapeutic Perspectives Giulia Ceglie 1,*et al of the same group in the discussion of the topic
Reviewer 2 Report
Simone et al. present interesting review entitled”Infantile brain tumors: a review of literature and future perspectives”. This review highlights the recent advances in the diagnosis and treatment of brain tumors in infants, with a particular focus on the molecular landscape and future clinical applications.It seems to be an important voice in clinical discussion concerning pediatric malignancies.
Major
figure 1
Please expand description concerning presented signaling pathways
Add epigenetic aspect of each group of tumors(miRNas interplay)if present.
cite:
Mazurek, M.; Grochowski, C.; Litak, J.; Osuchowska, I.; Maciejewski, R.; Kamieniak, P. Recent Trends of microRNA Significance in Pediatric Population Glioblastoma and Current Knowledge of Micro RNA Function in Glioblastoma Multiforme. Int. J. Mol. Sci.2020, 21, 3046. https://doi.org/10.3390/ijms21093046
And another study
https://doi.org/10.3171/2020.2.PEDS19715
Last 5 years references ratio does not exceed 50% (oscylates around 30%)Please update
Round 2
Reviewer 2 Report
Accept in current form.